# Applying Swin Architecture to Diverse Sign Language Datasets

**Yulia Kumar** *[ID], **Kuan Huang** *[ID], **Chin-Chien Lin, Annaliese Watson, J. Jenny Li, Patricia Morreale** [ID] **and Justin Delgado**

Department of Computer Science and Technology, Kean University, Union, NJ 07083, USA; watsanna@kean.edu (A.W.); pmorreal@kean.edu (P.M.); delgajus@kean.edu (J.D.)
* Correspondence: ykumar@kean.edu (Y.K.); khuang@kean.edu (K.H.)

**Abstract:** In an era where artificial intelligence (AI) bridges crucial communication gaps, this study extends AI's utility to American and Taiwan Sign Language (ASL and TSL) communities through advanced models like the hierarchical vision transformer with shifted windows (Swin). This research evaluates Swin's adaptability across sign languages, aiming for a universal platform for the unvoiced. Utilizing deep learning and transformer technologies, it has developed prototypes for ASL-to-English translation, supported by an educational framework to facilitate learning and comprehension, with the intention to include more languages in the future. This study highlights the efficacy of the Swin model, along with other models such as the vision transformer with deformable attention (DAT), ResNet-50, and VGG-16, in ASL recognition. The Swin model's accuracy across various datasets underscore its potential. Additionally, this research explores the challenges of balancing accuracy with the need for real-time, portable language recognition capabilities and introduces the use of cutting-edge transformer models like Swin, DAT, and video Swin transformers for diverse datasets in sign language recognition. This study explores the integration of multimodality and large language models (LLMs) to promote global inclusivity. Future efforts will focus on enhancing these models and expanding their linguistic reach, with an emphasis on real-time translation applications and educational frameworks. These achievements not only advance the technology of sign language recognition but also provide more effective communication tools for the deaf and hard-of-hearing community.

**Keywords:** Swin transformer; ASL detection; Taiwan Sign Language; deep learning; the unvoiced; multimodal large language models (MLLMs)

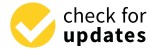



## 1. Introduction

In the contemporary digital era, characterized by the imperative for rapid and error-free communication, a significant discrepancy persists between verbal communication modalities and the communication necessities of individuals who rely on sign language, particularly within the deaf and hard-of-hearing communities [1,2]. This research endeavor seeks to address this communication divide by harnessing the capabilities of artificial intelligence (AI) and deep learning (DL) technologies. By focusing on American Sign Language (ASL) and Taiwan Sign Language (TSL) and conducting a comparative analysis of these, this study employs advanced convolutional and transformer-based neural networks for the recognition of images and videos from respective datasets. The technologies utilized in this research are considered to be among the top-tier methodologies, if not the leading approaches, for the accurate detection of sign language data.

This study predominantly emphasizes the hierarchical vision transformer using the shifted windows (Swin) model [3,4], alongside other models such as the vision transformer with deformable attention (DAT) [5], ResNet-50, and VGG-16 for comparative analysis. These models have demonstrated exceptional capabilities in accurately recognizing ASL. The primary dataset for image data training comprises Kaggle's ASL dataset, which includes 87,000 images, augmented with a project-specific, lightweight, and highly diverse dataset of approximately 300 images of the ASL alphabet. For video data, the model

training utilized 1200 short videos, evenly divided between TSL and ASL content. This study elucidates the differences between these datasets and the distinctions between the deployed DL models.

A pivotal component of this research is the development of two responsive applications designed for real-time ASL-to-English translation and vice versa. These applications are supported by an ASL educational framework that includes video lessons, a search function, and quizzes to evaluate learners' comprehension of the material. These applications, catering to diverse user groups, significantly contribute to fostering diversity, equity, and inclusion by assisting the integration of unvoiced individuals into society.

This research underscores the essential balance between accuracy, which necessitates considerable computational resources and time, and the requirement for real-time, portable language recognition capabilities, highlighting the trade-offs between these aspects. The employment of cutting-edge transformer models such as Swin, DAT, and video Swin transformers for diverse datasets, in comparison to the relatively rare TSL dataset [6] attached to this study, marks a novel approach in this field.

Furthermore, this research references recent studies that have utilized similar DL models for image classification [7,8] and leverages the researchers' experience to build upon these foundations. The introduction of the latest large language models (LLMs), including ChatGPT-4-Vision, Microsoft Copilot (preview, powered by DALL-E), and Google's Gemini 1.5, represents a new frontier in ASL recognition. This study discusses the current state of these models in recognizing and generating the ASL alphabet. Preliminary testing of the models, which must be continuous due to the sky-rocketing pace with which LLMs are being developed and released, demonstrates the superiority of ChatGPT-4-Vision, which is currently the latest and most advanced OpenAI model released to the public [9,10].

In the discussion section, the necessity of constant performance testing, as well as robust system testing, is debated, given the frequent upgrades to these technologies. This aspect is critical to ensuring the continuous improvement and reliability of ASL recognition systems [11].

Future work is projected to include fine-tuning LLMs like ChatGPT-4 and ChatGPT-4-Vision, access to which was kindly provided to the researchers by the Microsoft Research team on diverse sign language datasets, including those mentioned in this study.

In summary, the primary contributions of this study include the successful application of the Swin transformer technology for ASL and TSL recognition, the development of a real-time ASL-to-English translation prototype, and the exploration of the integration of multimodal data and large language models. These achievements not only advance the technology of sign language recognition but also provide more effective communication tools for the deaf and hard-of-hearing community.

## 2. Related Work

This endeavor to bridge the communication divide for the deaf and hard-of-hearing through artificial intelligence (AI) and deep learning (DL) methodologies spans continents and is rooted in a rich tapestry of global research efforts. These efforts have explored various aspects of sign language recognition, from hand gesture identification to the nuanced translation between sign languages and spoken languages. This expanded related work section provides a deeper analysis of each contribution within the context of the current study's objectives, highlighting the synergy between these foundational works and the innovative approaches being proposed.

Vashisth et al. [12] embarked on a significant endeavor to recognize Indian Sign Language using deep learning techniques, demonstrating the potential of neural networks in deciphering complex hand gestures. This study's importance lies in its focus on a less commonly researched sign language, offering valuable insights into the diversity and complexity of sign language recognition tasks. The study's methodology and findings contribute to a broader understanding of the challenges and opportunities in applying AI to sign language recognition across different linguistic and cultural contexts.

Alharthi and Alzahrani [13] further the discourse by investigating the efficacy of vision transformers coupled with transfer learning for Arabic Sign Language recognition. Their work underscores the adaptability and robustness of transformer models in handling the intricacies of sign language, providing a compelling case for the use of such advanced AI models in sign language recognition tasks. The cross-linguistic applicability of these models, as demonstrated in their study, enriches the technological toolkit available for bridging communication gaps across diverse sign languages.

Avina et al. [14] propose an AI-based framework specifically designed for translating American Sign Language (ASL) to English and vice versa. This framework is pivotal as it aligns closely with the current research's aim to develop responsive applications for real-time sign language translation. The methodologies and technologies employed in their study offer a blueprint for creating effective and inclusive communication tools that cater to the needs of the deaf and hard-of-hearing communities.

In a similar vein, De Coster and Dambre [15] explore the innovative use of pre-trained written language models for neural sign language translation. Their approach leverages the vast knowledge encapsulated in existing language models, applying it to the domain of sign language translation. This strategy not only exemplifies the potential for cross-domain application of AI technologies but also provides a methodological foundation upon which the current study can build, especially in terms of enhancing translation accuracy and efficiency.

Marzouk et al. [16] introduce an optimization technique—atom search optimization—combined with deep learning for Arabic Sign Language recognition. Their innovative approach to improving recognition accuracy by integrating evolutionary algorithms with neural networks offers a novel methodology that could enhance the effectiveness of the Swin model proposed in the current research for ASL and TSL recognition.

The practical applications of AI in sign language translation, as evidenced by the Brazilian Hand Talk mobile app and other English-to-ASL converters [17–23], showcase the tangible impacts of these technologies in enhancing communication accessibility.

The recent advancements in Swin transformers have led to significant developments across various fields. For instance, in the paper by Liang et al. [24], the focus is on image restoration, demonstrating the Swin transformer's effectiveness in enhancing image quality. Cao et al. [25] highlight its application in medical imaging, specifically in segmenting complex structures in medical images. In the realm of self-supervised learning, Xie et al.'s work [26] emphasizes the utility of Swin transformers in learning without labeled data, a major step forward in machine learning. He et al. [27] further apply this technology to remote-sensing images, improving semantic segmentation capabilities in geospatial analysis. Zu et al. [28] explore the use of Swin transformers for classifying pollen images, showcasing their potential in environmental and botanical studies. Nguyen et al.'s research [29] in dynamic semantic communication demonstrates the model's efficiency in handling diverse computational requirements in communication systems. The versatility of Swin transformers is further evidenced in MohanRajan et al.'s [30] study for land use and cover change detection and Ekanayake et al.'s work in MRI reconstruction [31], showing its effectiveness in both environmental monitoring and medical imaging.

In the field of video analysis, Lu et al. [32] apply Swin transformers to classify earthwork activities, enhancing the accuracy of such tasks. Lin et al.'s CSwinDoubleU-Net model [33] combines convolution and Swin transformer layers for improved medical image segmentation, particularly in detecting colorectal polyps. Moreover, Pan et al.'s study on renal incidentaloma detection [34] using a YOLOv4 + ASFF framework with Swin transformers marks a significant advancement in the detection and classification of medical conditions through imaging.

## 3. Methodology

This research utilizes the Swin transformer model, introduced by Liu et al. from Microsoft Research [4,35,36], for sign language recognition, focusing on American Sign Language (ASL) and Taiwanese Sign Language (TSL). The methodology is structured

into five main phases: diverse datasets used for this study, understanding of the Swin transformer, model training, results evaluation, and application development. Each phase is crucial for developing a robust system for sign language recognition and translation.

*3.1. Diverse Datasets*

The unvoiced community across the United States and Canada exhibits remarkable diversity, incorporating a wide array of dialects. This diversity includes the prevalent Black American Sign Language (BASL) and extends to dialects originating in Bolivia, Burundi, Costa Rica, Ghana, Nigeria, various Francophone regions, and Québec. Similarly, the United Kingdom and Australia each offer their unique versions of sign language, namely, British Sign Language (BSL) and Australian Sign Language (Auslan), respectively. These variations mirror the regional differences observed in spoken American English, where accents and colloquialisms can introduce communication barriers, even among speakers of the same language. In the context of ASL, although acoustic accents do not exist, variation manifests through distinct signs and gestures.

Figure 1 showcases the training dataset utilized in this study, drawing from Kaggle's extensive collection (displayed at the top of Figure 1a and bottom of Figure 1b) and a specially curated dataset for this research (shown at the bottom of Figure 1a and top of Figure 1b). This approach ensures a comprehensive understanding and representation of the diverse sign language dialects under consideration at a glance.

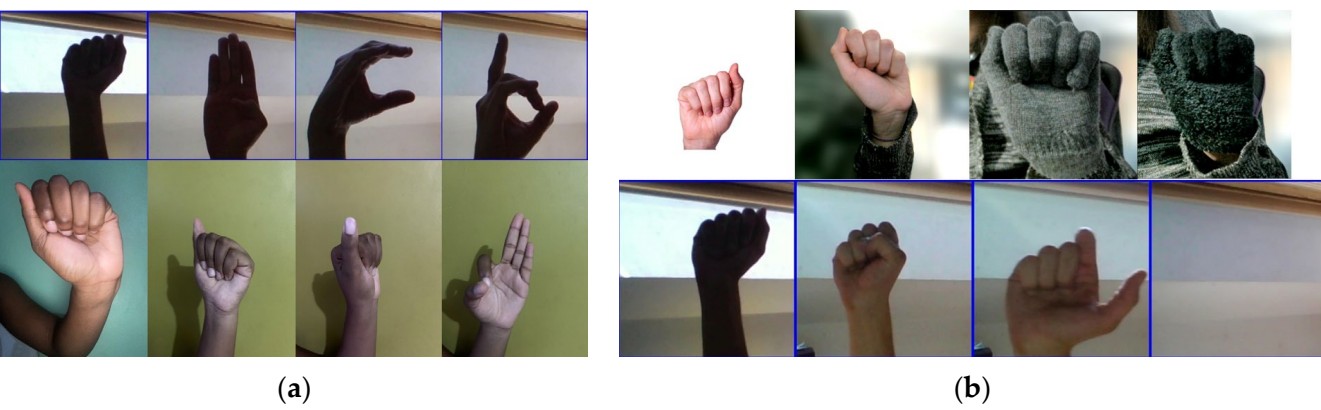

(**a**)             (**b**)

**Figure 1.** (**a**) Snapshots of Kaggle ASL dataset used for training (**top**) vs. lightweight custom dataset (**bottom**). (**b**) Projects test cases (**top**) vs. Kaggle data used for testing (**bottom**).

Figure 1 reveals that the initial training dataset predominantly features white male hands. To counter this limited representation, a supplementary dataset, curated by a female researcher proficient in ASL and non-binary person, introduces greater diversity. This step addresses a critical aspect of AI research in sign language recognition: the potential for algorithms to perpetuate biases. A diverse and representative dataset is imperative to ensure equitable recognition across different groups of signers, each with unique cultural backgrounds and signing styles.

In sign language, communication transcends mere hand movements to include facial expressions and body language. The nuances of these signs, shaped by varied expressions and postures, are pivotal for precise interpretation and must be accurately represented in the training data to prevent biases. Figure 2 illustrates the Taiwanese dataset employed in this study, featuring three phrases mirrored with the same phrases in ASL.

As was just mentioned, sign language communication comprises gestural changes and non-manual signals. Gestural changes denote different words through variations in hand shape, movement, location, and palm orientation, while non-manual signals encompass accompanying body movements and facial expressions [37,38]. Taiwanese Sign Language (TSL) traces its origins to Japanese Sign Language, reflecting influences from the Japanese colonial education system, and has evolved by integrating a significant portion of Chinese

Sign Language vocabulary [39]. In contrast, American Sign Language (ASL) emerged from French Sign Language and was further developed in the United States with the establishment of the first school for the deaf [40,41]. Despite sharing similar strategies, ASL and TSL differ in their components, leading to distinct representations. For example, ASL often uses the handshape of the first English letter to represent words. In TSL, "Family" is depicted by mimicking a roof and circling inside, whereas ASL represents it with a two-handed "F" handshape, with thumb-forefinger circles diverging and converging with the pinkies. Both ASL and TSL, like other sign languages, exhibit regional variations, with different locales developing unique vocabularies and dialects shaped by their linguistic environments. For instance, the sign for "know" in northern Taiwan involves a downward palm movement from the chest, whereas in southern Taiwan, it is indicated by tapping the chest with a clenched fist, highlighting the cultural evolution of local sign languages.

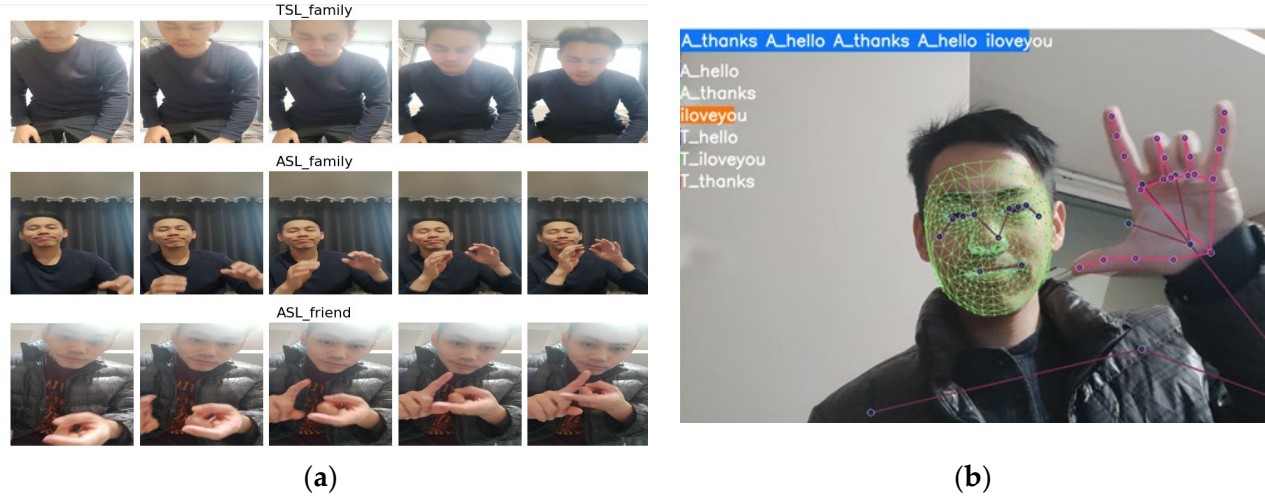

(a)  (b)

**Figure 2.** (**a**) Snapshot of the training dataset. (**b**) Live recognition of three phrases—"thanks", "hello", and "I love you"—in TSL vs. ASL. Colors indicate the probability of each sign.

The distinction between ASL and TSL is reflective of the broader diversity in sign languages globally. Effectively differentiating between various regional and national sign languages can enhance deaf education quality and foster integration and mutual understanding between the deaf and hearing communities. The dataset collection phase was crucial, involving the compilation of a diverse array of images and videos for training and evaluation. The primary ASL dataset consisted of 87,000 images from Kaggle [42], augmented by a curated collection of 300 images to increase demographic diversity and minimize algorithmic bias. A custom dataset featuring three phrases in TSL paralleled with ASL laid the groundwork for preliminary video recognition trials. Future efforts will extend to incorporating additional open-source datasets to enrich the training and testing corpus, accommodating the unique dialects within ASL and TSL.

Currently, the research focuses on experimenting with video recognition for ASL and TSL. Future work aims to explore other open-source datasets, including the ASL Lexicon Video Dataset, featuring over 3300 ASL signs [43]; the World Level American Sign Language Video Dataset on Kaggle, containing 12,000 processed videos [44]; ASL Citizen by Microsoft Research, a crowdsourced dataset with approximately 84,000 video recordings [45]; the MS-ASL Dataset, a large-scale collection of over 25,000 annotated videos [46]; the OpenASL Dataset, a comprehensive ASL–English dataset [47]; the How2Sign Dataset, a multimodal and Multiview continuous ASL dataset [48]; the YouTube-ASL Dataset, a large-scale corpus of ASL videos [49]; and the ASL video dataset—Boston University, featuring video sequences of distinct ASL signs [50].

Identifying reliable datasets for Taiwanese Sign Language presents challenges, yet resources like the Taiwanese Across Taiwan (TAT) Corpus and "a survey of sign language

in Taiwan" by SIL International offer insights into the language's dialects and regional variations [51].

### 3.2. Understanding Swin Transformer

The Swin transformer has become the subject of various projects [4], and its following iterations, including Swin transformer V2 and the video Swin transformer [35,36], signify significant progress in deep learning, especially in both image and video processing [32,52,53]. It has gained significant popularity due to its unique approach and effectiveness in various vision tasks. However, the approach of applying it to American Sign Language (ASL) and Taiwanese Sign Language (TSL), showcasing the model's versatility, is novel.

Introduced as a hierarchical transformer [4], the Swin transformer computes representations through shifted windows, facilitating efficient image processing across various scales. This versatile and robust architecture enables cross-window connections with minimal computational costs, allows for adjustment to different scales of visual data, and delivers superior performance across diverse benchmarks. Swin transformer V2 [35] built upon its predecessor, integrating refinements and optimizations to enhance both performance and efficiency. Preserving the foundational principles of the original Swin transformer, V2 focuses on augmenting the model's scalability, training stability, and efficiency. The video Swin transformer [36] extends the mentioned architecture to video content, aiming to capture both temporal dynamics and spatial hierarchies. It is particularly effective for videocentric applications, such as action recognition, video classification, and temporal segmentation. The video Swin transformer [36] introduced mechanisms to manage the temporal dimension of videos, potentially through 3D convolutions or specialized temporal windowing techniques. This capability enables the model to assimilate spatial and temporal features from video data effectively, offering a potent solution for analyzing complex video materials. The progression from the Swin transformer to Swin transformer V2, and subsequently to the video Swin transformer, illustrates an evolutionary path toward more sophisticated, efficient, and adaptable transformer models for visual data processing. Each iteration introduces enhancements or modifications that improve the model's suitability and performance for a wide array of tasks in image and video understanding.

The Table 1 below represents a summary of the Swin model, introduced by their creators.

**Table 1.** Summary of Swin transformers.

| Feature | Swin Transformer | Swin Transformer V2 | Video Swin Transformer |
|---|---|---|---|
| Domain | Image | Image | Video |
| Key Novelty | Shifted Windows | Scaling, Refinements | Spatiotemporal Attention |
| Compared to Swin | Base Model | Improved | Video Extension |

The uniqueness of this transformer is in its hybrid design that combines convolutional neural networks (CNNs) and transformers with features like shifted windows and hierarchical structure. These help it to handle the nuances of sign language across cultures. The most outstanding characteristics of the Swin transformer and its main competitors are presented in Figure 3.

Figure 3 illustrates the numerous advantages of the Swin transformer, a model renowned for its efficiency, scalability, and superior performance across various vision tasks. This study focuses on a comparison of the Swin transformer's performance against other models such as VGG-16, Resnet-50, and DAT [54]. The Swin transformer distinguishes itself by providing an optimal balance between efficiency and accuracy, particularly in scenarios requiring a comprehensive understanding of both local and global image features. It demonstrates exceptional ability in identifying and delineating multiple objects within images. Its application scope extends to semantic and instance segmentation, pose and depth estimation, transfer learning, and panoptic segmentation—a fusion of semantic and instance segmentation.

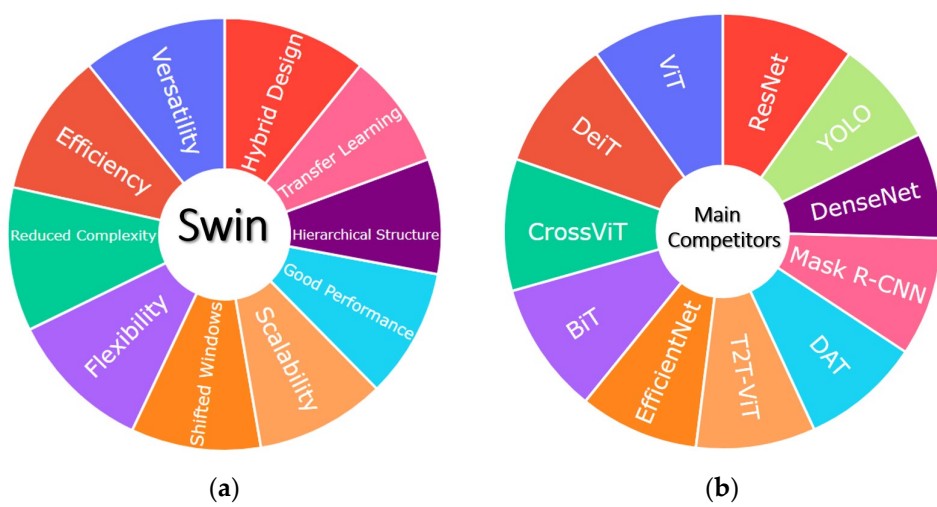

|  |  |
|---|---|
| (**a**) | (**b**) |

**Figure 3.** (**a**) Swin characteristics. (**b**) Main competitors of Swin transformer.

The Swin transformer's unique shifted window scheme, which processes images in non-overlapping windows with self-attention, followed by layer-wise window shifting for cross-window connections, enables it to efficiently encompass broader contexts and capture precise details crucial for sign language recognition. Unlike traditional transformers, Swin transformers create feature maps at various resolutions, enhancing their applicability in diverse vision tasks, including the nuances of hand gestures and facial expressions in sign language. Its versatility in processing images of different sizes and its hierarchical structure make it an advantageous tool for varied real-world applications, achieving state-of-the-art results in benchmarks like ImageNet and showing potential in sign language recognition by successfully telling apart similar signs with small differences. We include examples to show the effectiveness of the Swin transformer model. Later in the paper we demonstrate the Swin transformer model's effectiveness by presenting its classification accuracy, which reached 100% on the test dataset. Additionally, we illustrate the class activation maps of the Swin transformer, highlighting comprehensive image coverage in comparison to traditional convolutional networks and the DAT model. The pipeline of our methodology, from data collection to results evaluation, is illustrated in Figure 4.

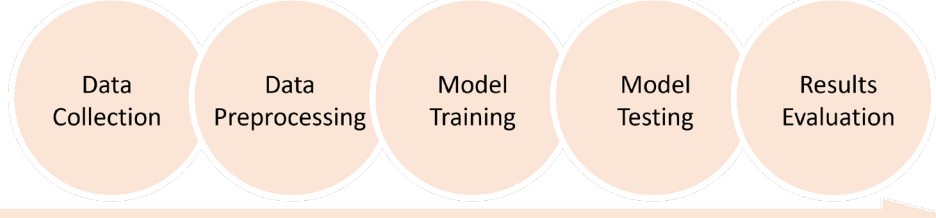

**Figure 4.** Swin transformer pipeline for sign language recognition.

### 3.3. Model Training

The researchers started their study with the original model but soon moved to its enhanced version, V2. To verify that Swin transformer V2 is particularly suited for sign language recognition, simulations were set up on Ubuntu 20.04.5 Linux system (Canonical Ltd., London, UK) with the following characteristics: AMD EPYC 7513 32-Core Processor 2.60 GHz (Advanced Micro Devices, Inc., Santa Clara, CA, USA) and 8 NVIDIA GeForce 3090 graphics cards (NVIDIA Corporation, Santa Clara, CA, USA), and each one has 24 Gigabyte memory. Details can be seen in Table 2.

The training took over 300 epochs and produced the following results (see Table 3).

**Table 2.** Simulation parameters.

| Trial Parameter | Comments |
|---|---|
| Initial Dataset | 87,000 images |
| Trial Dataset | 80% for training, 20% for testing at random |
| Classification | 29 classes (A to Z, Space, Del, and Nothing) |
| Batch Size | 16 |
| Trial Dataset | 256 × 256 (resized) |
| Optimizer used | SGD, learning rate 0.001 |
| Number of Epochs | 100 |
| Pytorch version | 1.12.1. |

**Table 3.** The model's characteristics and simulation results.

| Trial Parameter | Number of Parameters | Accuracy |
|---|---|---|
| DAT Transformer | 86,886,357 | 99.99% |
| VGG-16 | 165,845,085 | 100% |
| ResNet-50 | 23,567,453 | 100% |
| Swin transformer | 65,960,349 | 100% |

Table 3 presents a comparative analysis of deep learning models in terms of their parameters and accuracy in ASL recognition. The table highlights ResNet-50 as the most parameter-efficient model, making it potentially more suitable for mobile applications. In contrast, VGG-16, with the highest parameter count due to its three fully connected layers, may be less optimal for such applications. Despite this, all models, including the DAT and Swin transformers, achieved high accuracy, illustrating a balance between model complexity and performance in ASL translation tasks. Further insights into these models' performance are provided through training and testing loss curves in subsequent visualizations. It was found that the DAT transformer did not outperform the Swin transformer in this project, which does not match the original paper of on the DAT transformer [54] that claimed that it should. The confusion matrix for VGG-16, ResNet-50, and the Swin transformer, of which achieved a 100% accuracy rate, is depicted in Figure 5a.

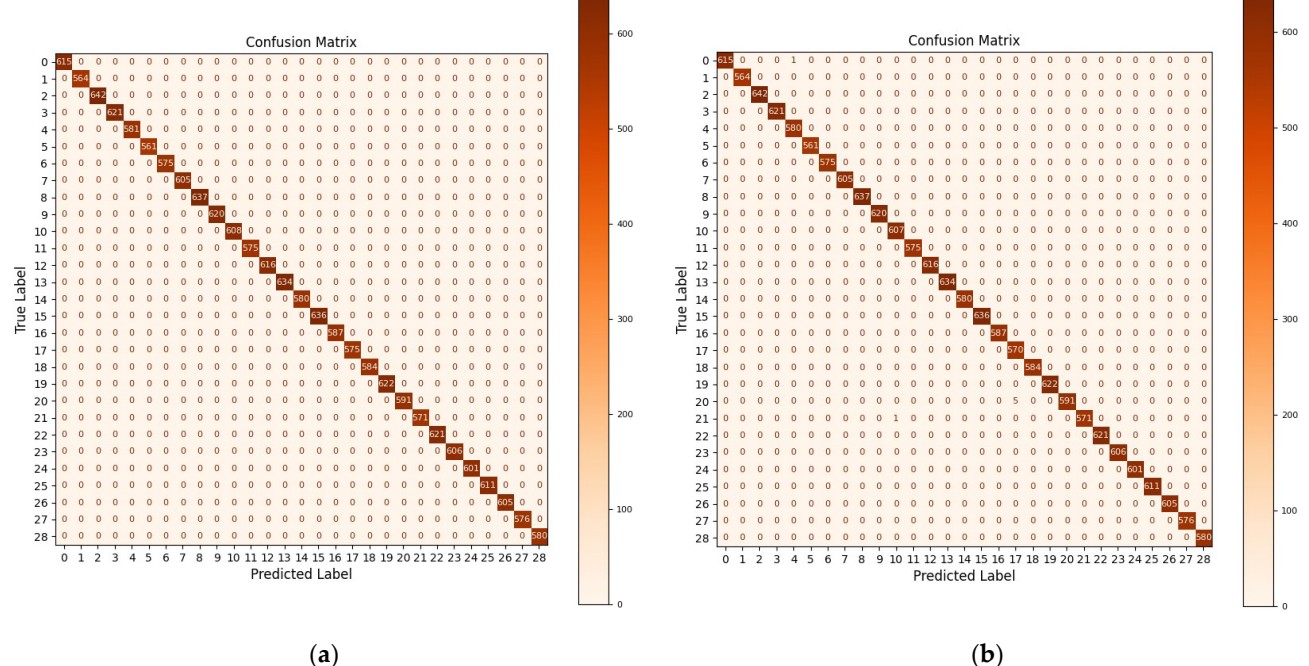

(**a**)　　　　　　　　　　　　　　　　　(**b**)

**Figure 5.** (**a**) Confusion matrix for three leading models (achieved 100%). (**b**) Confusion matrix for DAT transformer (achieved 99.99%).

Figure 6 represents the training and testing loss curves for all models.

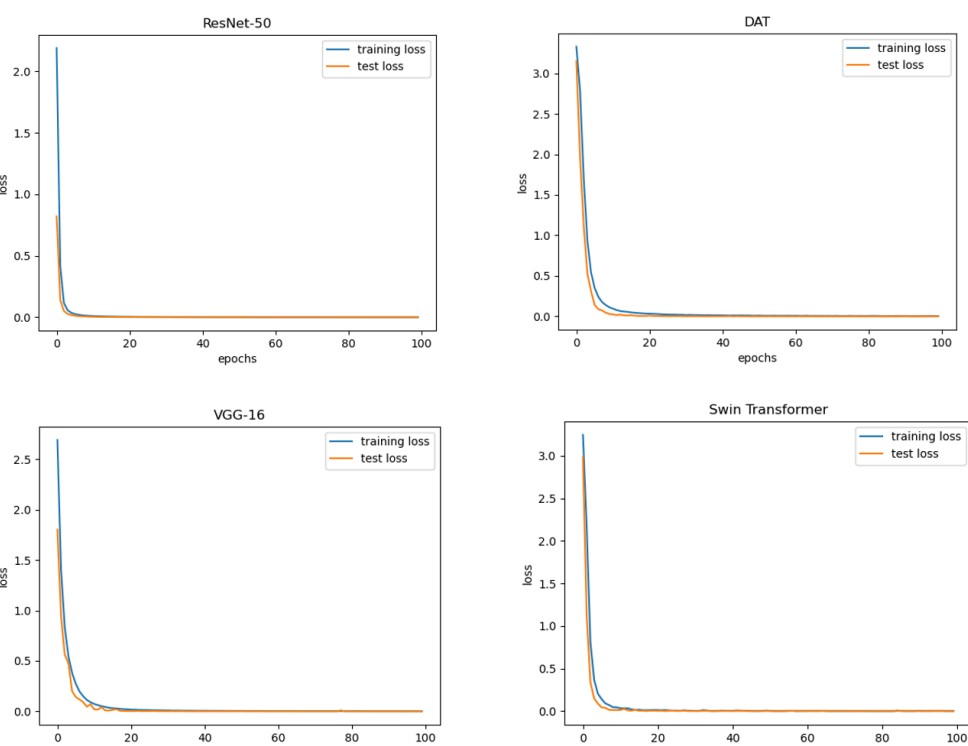

**Figure 6.** Training and testing loss curves of the VGG-16 (**bottom left**), ResNet-50 (**top left**), DAT Transformer (**top right**), and Swin transformer (**bottom right**).

As can be seen from Figure 6, all four models demonstrated huge a reduction in losses right away, happening around less than 10–15 epochs. To further understand the model, the researchers then proceeded with bias analyses. The analysis was built upon previously developed strategies and applied to Meta's DETR transformer families [55,56].

The visualization of the biases discovered in the models can be seen in Figure 7.

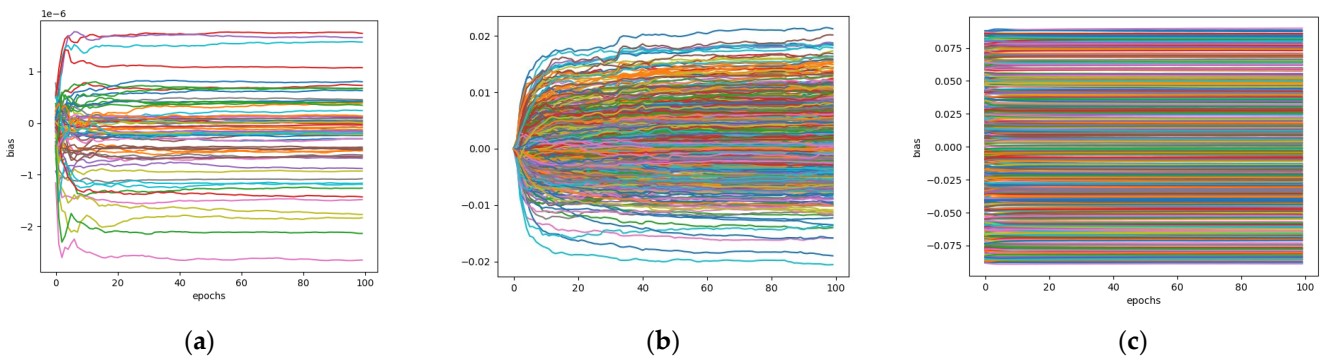

**Figure 7.** Bias Visualizations, each colored line represents the bias value changes of a single neuron during training: (**a**) The first convolutional layer of VGG-16. (**b**) The multilayer perceptron of the first transformer stage of the Swin transformer. (**c**) The multilayer perceptron of the first transformer stage of the DAT Transformer.

According to Figure 7, the VGG-16 model demonstrates stable bias values after 15–20 epochs, within a range from $-2.75$ to $1.75$ ($1 \times 10^{-6}$). For the Swin and DAT transformers, the focus is on the first transformer stage's multilayer perceptron (MLP) biases [57]. The Swin transformer shows a unique dome-like bias shape, suggesting a need for deeper analysis in terms of distribution, density, and outliers. Conversely, the

DAT transformer's biases converge around epochs 45–50 then stabilize, indicating less fluctuation post-convergence. This analysis aids in understanding the learning behaviors of these models.

The ResNet-50 CNN model demonstrated no biases in its first convolutional layer. SParameters of the model can be seen from (1):

$$self.conv1 = nn.Conv2d(self.inplanes, 64, kernel\_size = 7, stride = 2, padding = 3, bias = False) \tag{1}$$

The ResNet-50 model's first convolutional layer is designed without bias parameters to streamline the number of variables and potentially improve computational efficiency. This decision was made since in deep learning models, especially in convolutional neural networks, bias terms are sometimes omitted. This is because batch normalization, often applied after convolutional layers, negates the effect of the bias by standardizing the output. Therefore, removing the bias parameter can reduce the model's complexity without significantly affecting its performance. AI biases and explainable AI are at the forefront of Artificial Intelligence research due to their importance in ensuring AI models are used responsibly.

Further in this section, class activation maps (CAMs) are employed to visually interpret the focus areas of deep learning models used in image classification, which is vital for understanding the decision-making process of AI [58].

As can be seen from Figure 8, CAMs produced heatmaps that identify critical regions, influencing the classification decision and offering a comprehensive view when combined with the bias data. This is particularly useful in explicating the opaque decision-making process in deep learning, enhancing user trust by demystifying AI classifications. The figure highlights the model's attention to various features that affect accuracy.

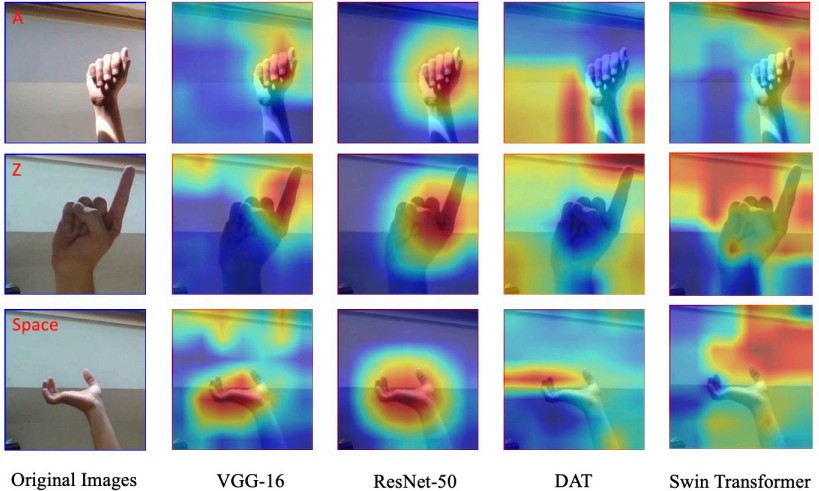

| Original Images | VGG-16 | ResNet-50 | DAT | Swin Transformer |

**Figure 8.** The CAMs of the Swin transformer in comparison with other DL models. where colors highlight focus areas—warm for higher attention.

### 3.4. ASL vs. TSL Video Recognition

Currently, only preliminary results of ASL vs. TSL video classification are available. The training started with training the recurrent neural network (RNN) and six classes of video classification mentioned above and demonstrated in Figure 9b.

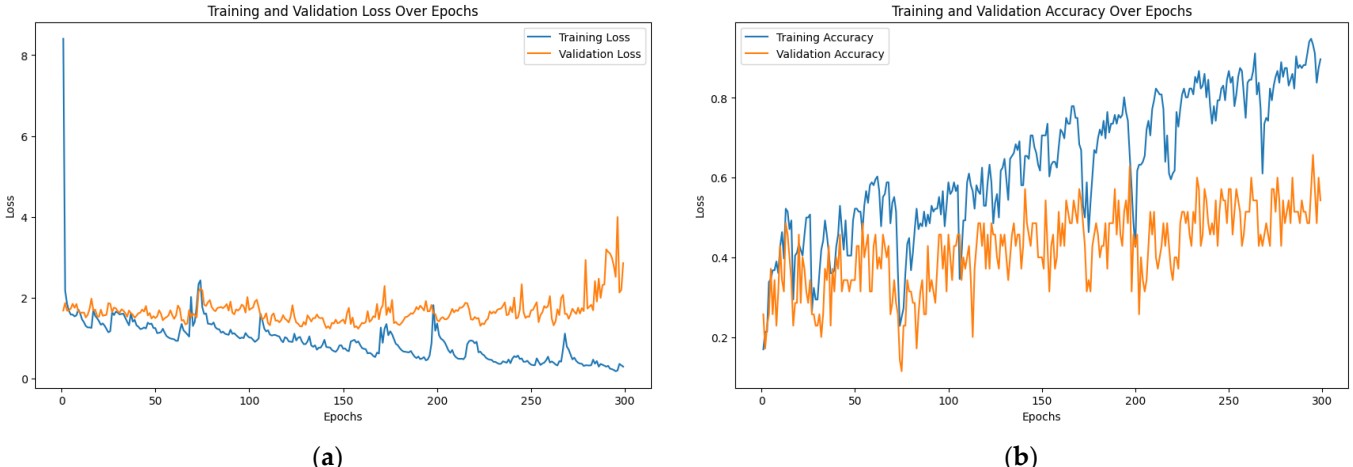

(**a**)　　　　　　　　　　　　　　　(**b**)

**Figure 9.** (**a**) Training and testing losses of ASL vs. TSL video classification with RNN. (**b**) Training and testing accuracy of ASL vs. TSL video classification with RNN.

As can be seen from Figure 9b, the training accuracy achieves nearly 100% at 300 epochs, which is well performed considering a lightweight dataset of study. However, the poorly performing validation accuracy and associated loss (the orange curves) demonstrated the model overfit, forcing researchers to reduce the learning rate and try various optimizers other than Adam. It was also decided to enrich the training dataset by adding to original phrases words "family", "friends", and "learn" in both ASL and TSL. An extra folder with "ping pong" in TSL was also used. Updated training and testing datasets can be found in Figure 10.

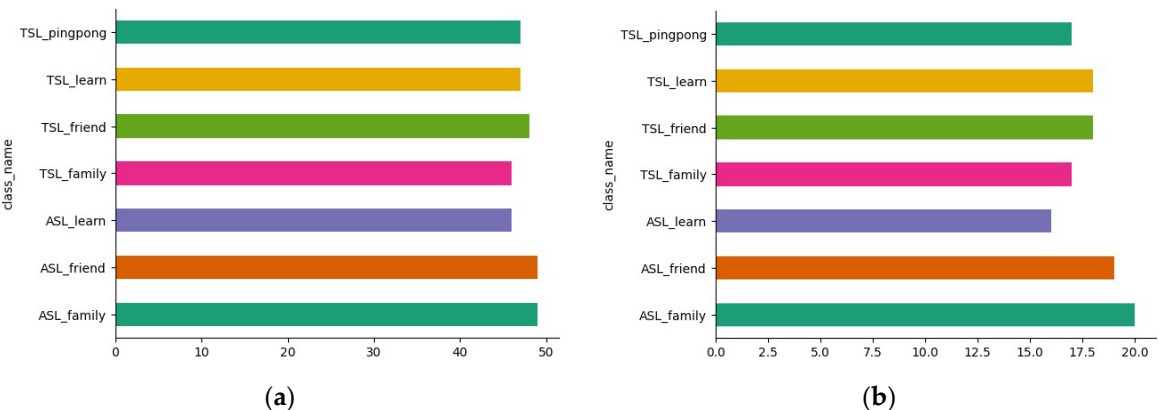

(**a**)　　　　　　　　　　　　　　　(**b**)

**Figure 10.** (**a**) Training vs. (**b**) Testing. ASL and TSL datasets.

The Adam optimizer, an extension of the Adam optimization algorithm, was used for this training. This optimizer features weight decay for regularization, a technique that helps prevent overfitting by penalizing large weights. The learning rate was set to 0.001. This crucial hyperparameter controls the step size at each iteration while moving toward a minimum of the loss function. A well-chosen learning rate will help the model to converge to a good solution efficiently. The weight decay was set to 0.0001. Weight decay is a form of regularization that adds a small penalty to the loss function for larger weights, encouraging the model to learn simpler patterns and potentially reducing overfitting. To understand the structure of the transformer, researchers visualized the architecture of the first two layers of the video Swin transformer model dynamically. The results can be seen in Figure 11.

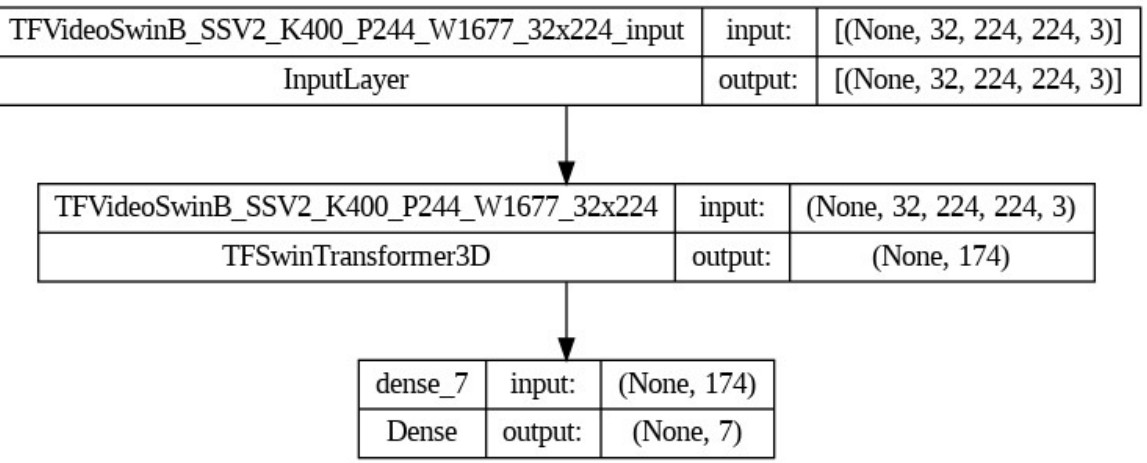

**Figure 11.** Architecture of the video Swin transformer.

As can be seen in Figure 11, the schematic representation of a Swin architecture consists of the input layer, namely, a 4-dimensional tensor with the shape of 32 frames in each video input, 224 × 224 spatial dimensions of each frame, and RGB number of color channels in each frame. A pre-trained video Swin transformer model with specific configuration and weights TFVideoSwinB_SSV2_K400_P244_W1677_32x224 is designed to handle video input and perform 3D convolutions or other spatiotemporal feature extraction operations. Its input and output shapes indicate that the model maintains 32 frames and 224 × 224 spatial dimensions but reduces the channel dimension from 3 (RGB) to 174. This reduction suggests that the model extracts 174 features from each frame. The dense, fully connected layer takes the output from the previous Swin transformer layer and produces an output shape, see Table 4 for layer details. The model from Figure 11 performs a classification task with seven classes. The dense layer is responsible for mapping the extracted features to the probabilities of the classes.

**Table 4.** Video Swin model characteristics.

| Layer (Type) | Number of Parameters | Output Shape |
|---|---|---|
| TFVideoSwinB_SSV2_K400_P244_W1677_32x224 (TFSwinTransformer3D) | 88,834,038 | (None, 174) |
| dense (Dense) | 1225 | (None, 7) |

Training the model on Google Collab was extremely challenging as the GPU capacity for personal use is very limited. Figure 12 represents the bottlenecks.

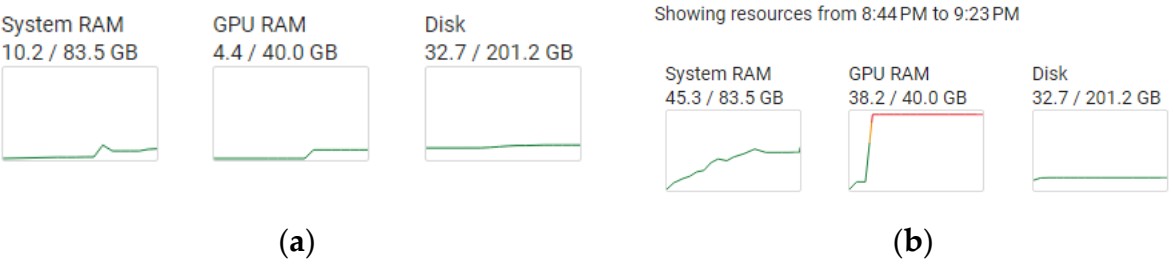

**(a)**    **(b)**

**Figure 12.** Collab Pro+ resources (**a**) before and (**b**) during the training. The color represent the gpu usage levels.

Preliminary results of applying the Swin video transformer on ASL vs. TSL datasets can be seen in Figure 13.

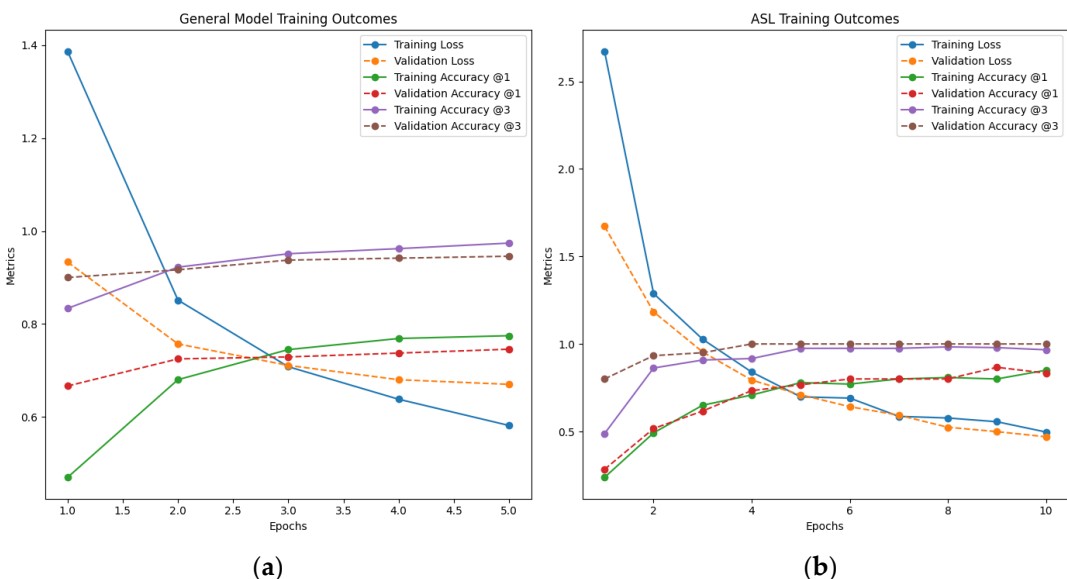

(**a**)　　　　　　　　　　　　　　　　　　　(**b**)

**Figure 13.** (**a**) Training and testing losses of ASL vs. TSL video classification with video Swin transformer. (**b**) Training and testing accuracy of ASL vs. TSL video classification with video Swin transformer.

### 3.5. Application Development

This research tackles the communication problem between those who know and use ASL every day and those who do not. In simple words, the researchers aim to create a swift means of understanding by providing smooth communication for those involved. Case studies and associated applications consist of two types and associated cases:

(1) Develop a user-friendly interface for ASL translation, ensuring it is suitable for the intended users and use cases.
(2) Create interactive and engaging learning tools for ASL education.

To address the first case, two different applications, the Smooth Talk app and the STApp app, were created. Both feature a very simple and user-friendly interface. The goal was originally to develop only one application, and STApp stands for Smooth Talk app as well, but eventually, two different apps were developed by the team of researchers. Both apps capture the ASL language live and translate it into English using a Python backend. The first yellowish version of the app was inspired by the low-code AI Teachable Machines web tool [59], which was used to practice ASL language. Figure 14b (bottom) demonstrates an accuracy of 94% for the letter "B", which was the top accuracy that could be achieved with the app as it was built to use a custom light ASL dataset collected for the project. As can be seen in Figure 14a, another prototype demonstrates an accuracy of 70.14%. Figure 14c demonstrates the live demo of the Smooth Talk app.

The mobile-first GUI of both apps relies on the Bootstrap framework, CSS flex, and other front-end technologies targeting responsive web design to further accommodate the users and potentially allow them to use the app from any place. They can both be considered hybrid apps currently working equally well both on a smartphone and in the web browser. The use of JavaScript and its APIs, for example, to convert text to speech and deliver it to the hearing side of the conversation further enhances the prototype. The speech is converted to text using the same API. The text is translated into ASL letters using Map, aka dictionary data structure, and the result is displayed to the unvoiced person. The look and feel of the Small Talk app resemble the website of NAD Youth [2]—a project of the National Association of the Deaf [1]. It also inspired the STApp app GUI.

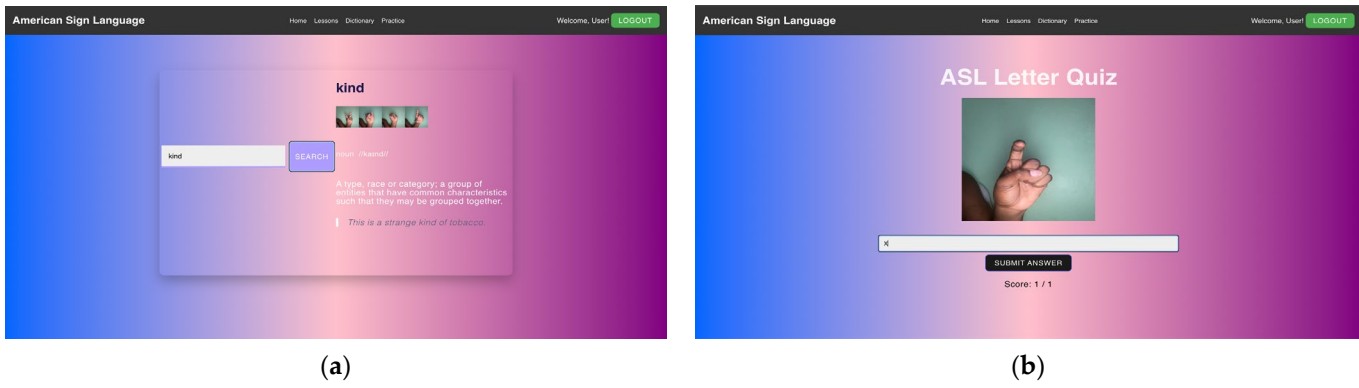

(**a**)  (**b**)  (**c**)

**Figure 14.** (**a**) Snapshot of the STApp app pages. (**b**) Homepage of the NAD Youth website [2] (**top**) vs. the Smooth Talk homepage (**middle**) vs. ASL recognition page of the app (**bottom**). (**c**) Live demo of the app.

As can be seen from Figure 14a, the STApp app uses emojis as icons on its buttons, which adds uniqueness to this web design. The STApp app also provides the functionality of taking a selfie and prerecording a video; then, it sends it to the other side of the conversation, accommodating asynchronous communication and making it aware of the translation accuracy.

To address the second educational case, another app was developed (see Figure 15).

(**a**)  (**b**)

**Figure 15.** (**a**) Snapshot of the educational app ASL translation page. (**b**) Quiz page of the app.

At this point, the app user can log in to the educational system, watch video recordings of the ASL language lesson, and take the quiz to test their knowledge. The researchers are considering converting this quiz into an app game to add interactivity to it and attract a broader population. Obviously, the Swin transformer and other AI models and tools can be used in ASL educational platforms for improved learning experiences and interactive applications, or games can further assist with that. Eventually, the researchers plan on integrating it into the platform for both ASL and TSL. Once the research project was launched, the researchers themselves had to learn the basics of the ASL language to some degree.

## 4. LLM Integration

Integrating large language models (LLMs) into this research can enhance interpretability and user interaction with the system. Approaches such as automated annotation with LLMs are becoming mainstream. LLMs can generate descriptions or labels for video clips or images based on predefined criteria, which can then be verified or refined by experts. As LLMs are constantly improving, they will eventually be able to assist in identifying potential gaps or biases, as it is critical to avoid biases in ASL recognition.

Sign language recognition and translation is a complex task that involves understanding subtle hand movements, facial expressions, and body language. It is also hard to find researchers or enthusiasts fluent in sign language and willing to devote their time and efforts to such a project. Sign language is both spatial (involving the positioning and movement of hands, fingers, facial expressions, etc.) and temporal (the meaning can depend on the sequence of movements). The researchers conclude that the Swin transformer's capabilities could potentially be very relevant to this dataset as the shifted window approach could efficiently handle the spatial aspect of sign language. Its ability to handle video data (spatial–temporal) makes it suitable for interpreting sign language in a dynamic, continuous context. Integrating large language models (LLMs) like GPT-4+ or similar technologies can further enhance the capabilities of the proposed apps. LLMs are expected to be particularly useful in ASL education and to facilitate interactive language learning experiences. In data classification, they can assist in interpreting and summarizing classification results. LLMs can provide contextual translations of signs or offer cultural insights into sign language usage, help practicing sign language, or explain concepts.

The latest LLMs, such as ChatGPT-4-Vision and Gemini, were tested on the ASL dataset. Special AI assistant, aka custom GPT Sign Speak Guide, was created with the help of OpenAI API. It can be seen below in Figure 16:

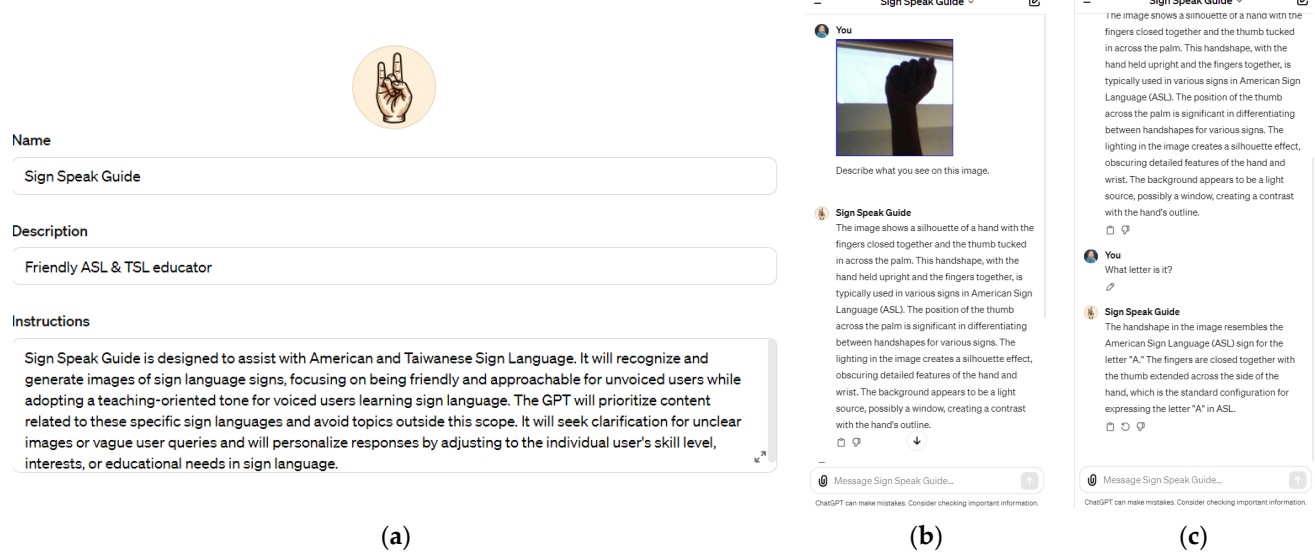

**(a)**        **(b)**        **(c)**

**Figure 16. (a)** The snapshot of the AI assistant Sign Speak Guide in the process of ASL recognition. **(b,c)** The configuration snapshot of the AI assistant Sign Speak Guide.

The assistant that utilizes the ChatGPT-4-Vision model can work with both ASL and TSL, but the accuracy of ASL is slightly better. Figure 16 demonstrates an image of the letter "A" fed to the bot. As can be seen from Figure 16, the AI assistant correctly recognizes the letter "A" provided.

As can be seen from Figure 17, neither of the leading models successfully generated it. The models were able to comprehend the meaning of ASL generation but could not correctly address the request.

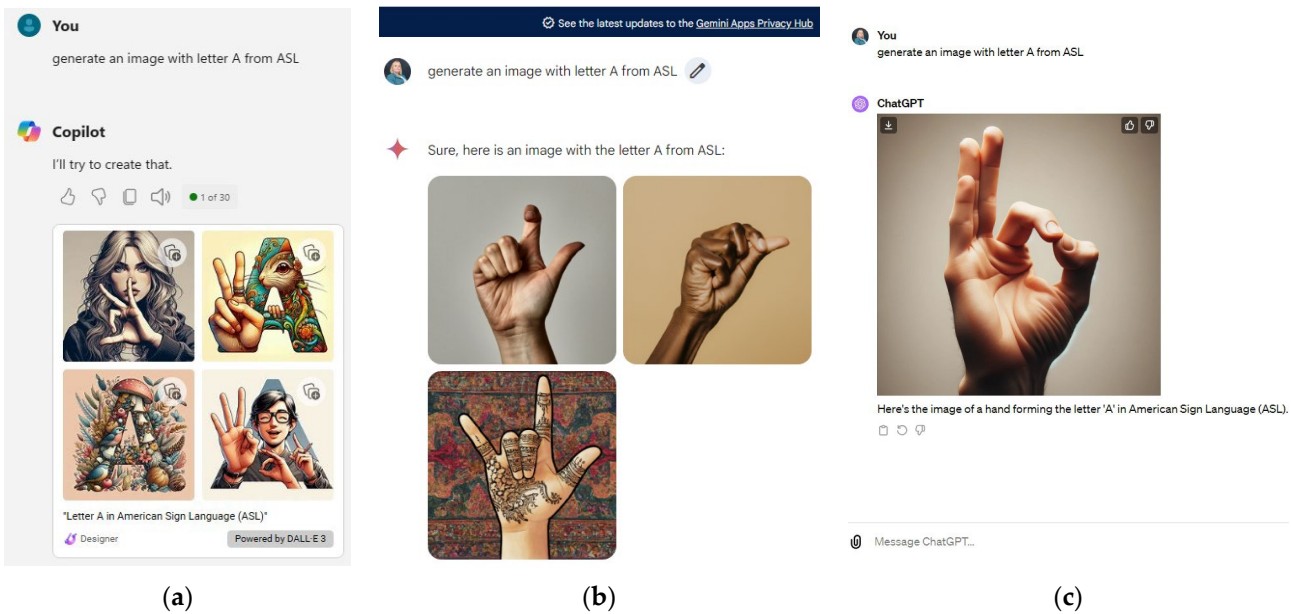

(**a**)          (**b**)          (**c**)

**Figure 17.** (**a**–**c**) The snapshot of ASL recognition by major LLMs.

Integrating multimodality into the research on ASL dataset classification, especially when combined with large language models (LLMs), can create a more comprehensive and effective system. Integrating not only images and text but also audio and video will improve understanding and interaction. The top standard would be creating an interface that adapts to the user's preferences and accessibility needs.

## 5. Conclusions and Future Work

The research and developed app prototypes will facilitate communication between those who primarily communicate through sign language and those who do not. Our trials resulted in the following accurate outcomes: the Swin transformer achieved 100%, and CNN models achieved 100% as well. However, we acknowledge that the dataset utilized is relatively simple, with most of the models achieving near-perfect accuracy on the test dataset. We plan to use more diverse datasets (mentioned in Section 3.1) for training and evaluation in the future works. Future research will include exhaustive testing of the prototypes and LLMs in the field of ASL recognition. It is expected that validation will make a reliable ASL tool for all possible. The ethical scope of the problem we tackle is very sensitive, and the handling of personal data must be discussed.

The emergence of comprehensive datasets has been instrumental in the development and testing of advanced sign language recognition models. The video Swin transformer [3], with its potential in video-based sign language recognition, represents a new era in understanding and interpreting sign language through visual data.

Despite these advancements, challenges remain. One of the primary challenges is the creation of large, diverse, and high-quality datasets that accurately represent the complexity of ASL and TSL. Additionally, real-time processing capabilities are crucial for practical applications of these technologies. While our models achieved high levels of accuracy, the diversity and representativeness of the datasets used may limit their generalizability.

Balancing accuracy with the need for real-time, portable language recognition capabilities remains a challenge for practical applications. Future research should focus on tailoring transformer and LLM-based models to accommodate the specific requirements of sign language recognition more effectively. Bridging the gap between academic research and practical, real-world applications of ASL and TSL detection technologies is essential.

In conclusion, the integration of transformers and LLMs in ASL and TSL detection represents a significant advancement in the field. These technologies offer enhanced capabilities for interpreting sign language, leading to more accurate and efficient ASL and TSL detection systems. However, continuous efforts in data enhancement and model optimization are crucial to address existing challenges and further advance the field. The datasets and models discussed herein offer a glimpse into the current state of the field and its potential trajectory. Continuous advancements in this domain hold the promise of bridging communication gaps for the deaf and hard-of-hearing communities globally, enhancing inclusivity and accessibility. With future improvements in Swin transformers and LLMs, the systems must be refined and updated to remain cutting-edge apps.

Soon, research will fine-tune LLMs like ChatGPT-4-Vision on both their public and diverse custom ASL datasets. It is also projected that the usage of high-performance computing will improve the parameters of the Swin transformer. This research contributes significantly to the field by bridging the communication gap for the deaf and hard-of-hearing communities through innovative AI and DL applications. It opens new pathways for the integration and inclusion of unvoiced individuals in society, marking a step forward in the quest for universal communication accessibility.

In the realm of future work, our focus will be placed on further integrating LLMs into sign language recognition systems. We plan to explore the use of advanced LLMs such as GPT-4 to enhance semantic analysis within these systems. Specifically, future research will investigate how LLMs can be utilized to improve the interpretation of complex sign language constructs and to generate more accurate natural language translations of sign language expressions. This integration is expected to challenge traditional methods and establish new benchmarks by enhancing semantic analysis and improving recognition accuracy. Additionally, we aim to examine the potential of LLMs in facilitating more interactive and engaging sign language learning experiences, as well as in providing insights into the cultural nuances of sign language usage across different regions. The integration of LLMs holds the promise of significantly advancing the capabilities of sign language recognition technology, thereby contributing to the creation of more inclusive and accessible communication tools for the deaf and hard-of-hearing communities.

**Author Contributions:** Conceptualization, Y.K. and K.H.; methodology, Y.K. and K.H.; software, J.D., C.-C.L. and A.W.; validation, C.-C.L., Y.K., K.H. and J.J.L.; formal analysis, J.D., C.-C.L. and A.W.; investigation, Y.K., J.D., C.-C.L. and K.H.; resources, Y.K.; data curation, P.M.; writing—original draft preparation, Y.K. and K.H.; writing—review and editing, J.J.L. and P.M.; visualization, Y.K. and K.H.; supervision, P.M.; project administration, Y.K.; funding acquisition, Y.K. All authors have read and agreed to the published version of the manuscript.

**Funding:** This research was funded by the NSF, grants awards 1834620 and 2137791, and Kean University's Students Partnering with Faculty 2023 Summer Research Program (SPF).

**Data Availability Statement:** Data are available on request due to privacy restrictions (i.e., the personal nature of ASL communication).

**Conflicts of Interest:** The authors declare no conflicts of interest. The funders had no role in the design of this study, in the collection, analyses, or interpretation of data, in the writing of the manuscript, or in the decision to publish the results.

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
