# Peer review of "Applying Swin Architecture to Diverse Sign Language Datasets"

_electronics, doi:10.3390/electronics13081509_

Round 1

Reviewer 1 Report

Comments and Suggestions for Authors

In this paper, the author propose a swin transformer based method for sign language. Equipped with the shifted window mechanism, the transformer is able to deal with nuances of sign language across cultures. Moreover, diverse datasets are collected for the experiments. To evaluate the performance of their method, they compare the method with other baselines. The topic is quite innovative. I suggest the author to make a minor revision. After reading this paper, I have following suggestions:

1st  The author claimed that shifted window mechanism combined with transformer can help to handle the nuances of sign language across diverse language in section 3.2. However, they only introduce use others summary and didnt explain why it works in their task.  I suggest the author to give an in-depth analysis for the advantage of using Swin in their task. If possible, please add some examples to show the effectiveness of the proposed network.

2nd  In 3.1, the author introduce the procedure of collecting diverse data and in 3.2 introduce the method. I suggest the author give a pipeline plot.

3rd  I suggest them to add an ablation study to verify the robustness of their algorithm given an noisy environment.

4th I suggest them to highlight their main contribution at the end of first paragraph.

5th In section 4, the proposal for future work by integrating LLM is interesting. Please add some concrete plans for the task.

6 Pay attention to the use the accurate terminology.

Comments on the Quality of English Language

Minor error, need to polish.

Author Response

Please find our responses attached.

Reviewer 2 Report

Comments and Suggestions for Authors

This study presented the application results for the American and Taiwan Sign Language communities through AI models of Swin architecture (it is described as the hierarchical vision transformer with shifted windows). That is, the presented research shows the Swin’s adaptability across sign languages, aiming for a universal platform for the unvoiced. This paper is evaluated as having limitations in the research results in that it does not present a new artificial intelligence algorithm. Therefore, the reviewer recommends that the following contents will be additionally revised.

  1. There is a lack of explanation for the contribution of the study compared to other research results.

  2. There seems to be a limit to the analysis results of English and Taiwanese languages.

  3. A more detailed description of the abstract of the study should be provided.

  4. The limitations of the research and its results must be addressed to the Conclusions.

  5. It is necessary to review how to apply the algorithm in the BERT model, which is known as an excellent text mining technique recently if it is possible to apply.

  6. In addition to the deep learning model, comparison and analysis results with excellent results for recent sign language recognition should be added or explained.

Comments on the Quality of English Language

Minor corrections are needed.

Author Response

Please find our responses attached. We adjusted our paper according to your valuable comments.

Round 2

Reviewer 2 Report

Comments and Suggestions for Authors

Reviewer checked the author's revised opinion.

Finally, reviewer would like you to check the article writing rules.

Truly yours.